# The Accidental Philosopher and One of the Hardest Problems in the World

**Sonje Finnestad and Eric Neufeld ***

Department of Computer Science, University of Saskatchewan, Saskatoon, SK S7N 5A2, Canada; sonje.f@usask.ca
* Correspondence: eric.neufeld@usask.ca

**Abstract:** Given the difficulties of defining "machine" and "think", Turing proposed to replace the question "Can machines think?" with a proxy: how well can an agent engage in sustained conversation with a human? Though Turing neither described himself as a philosopher nor published much on philosophical matters, his Imitation Game has stood the test of time. Most understood at that time that success would not come easy, but few would have guessed just how difficult engaging in ordinary conversation would turn out to be. Despite the proliferation of language processing tools, we have seen little progress towards doing well at the Imitation Game. Had Turing instead suggested ability at games or even translation as a proxy for intelligence, his paper might have been forgotten. We argue that these and related problems are amenable to mechanical, though sophisticated, formal techniques. Turing appears to have taken care to select sustained, productive conversation and that alone as his proxy. Even simple conversation challenges a machine to engage in the rich practice of human discourse in all its generality and variety.

**Keywords:** Imitation Game; Turing Test; computational linguistics; artificial intelligence



## 1. Introduction

When Turing [1] (p. 433) wrote about the difficulty of answering the question "Can machines think?", he stated that "If the meaning of the words 'machine' and 'think' are to be found by examining how they are commonly used it is difficult to escape the conclusion that the meaning and the answer to the question, 'Can machines think?' is to be sought in a statistical survey such as a Gallup poll", which he rightly described as absurd. Instead, he proposed a proxy called the Imitation Game wherein a machine would engage in conversation with a rigorous human interrogator with the aim of being judged human. Doing well at this seemingly singularly human practice would sidestep the difficult tasks of defining intelligence, human or machine, and would replace the target question.

From Turing's 1950 paper, which provides sample snippets of conversation between an interrogator and an imagined machine (which he calls the Witness) mimicking a human, it is clear that Turing intended that for a machine to be judged human, it needed to directly respond to tough questions from a demanding interrogator and to sustain, engage in, and contribute collaboratively to the dialogue.

The Imitation Game came to be known as the Turing Test, but for many the term "Turing Test" has become confused with Turing's [1] (p. 442) performance predictions for computer conversation by the end of the 20th century. He expected that by then an average interrogator after five minutes of questioning would have not more than a 70 percent chance of making a correct identification. Thus, after a 2014 Turing Test competition [2] in which more than 30 percent of human judges considered an unfocussed chatbot called Eugene Goostman to be human, event organizers claimed the test had been passed. This news spread quickly through the mainstream media.

Critics [3,4] pointed out this error, which was reflected in the media. Some scholars used the Goostman episode to advocate for more focused tests [5]; some echoed earlier

concerns proposing less stringent tests [6]. In fairness to all concerned, Turing's prediction about the performance of turn-of-the-century chatbots was remarkably accurate and arguably suggested that he may have considered the stated level of performance a milestone towards eventual success.

Our response was to examine Turing's original words to pin down the meaning of his test. In the course of that project, we revisited a shift in Turing's thinking between 1948 and 1950, which Hodges [7] (p. 296) said "boldly extended the range of 'intelligent machinery' to general conversation", whereas we consider it a seismic shift that displaced Turing's 1948 "branches of thought" with unrestricted human-level conversation (the Imitation Game) as the proxy for testing whether machines could exhibit human-level intelligence. This has become relevant now (2022) because the last decade has witnessed supra-human performance by software in several of the very domains Turing named in 1948 (e.g., chess and translation, but also Go and real-time text-to-speech conversion), perhaps suggesting that computers capable of unrestricted conversation are on the horizon. For the time being, however, the needle has barely moved on the Imitation Game.

Thus, the next section elaborates the preceding observations, and the two subsequent sections go on to discuss how and why several of Turing's original branches of thought (in particular, games and certain language recognition/transformation problems) have been found to be amenable to automation. Then, we then ask whether and what Turing might have understood about the Imitation Game, as well as games and language processing, that made conversation qualitatively more difficult to automate than the other branches of thought—so much so that he did not refer to them in 1950.

To clarify the terminology and assumptions made herein, we use *Imitation Game* to refer to the test as proposed by Turing [1] and *Turing Test* to describe chatbot competitions and the like. Scholars interpret the Imitation Game differently [8–10]; we interpret the Game as a competition between a human and a machine programmed to be judged human by a rigorous interrogator. We remind the reader that the *Mind* paper does not use phrases such as "*pass* the Turing Test" or "*pass* the Imitation Game", but instead uses terms like [1] (p. 422) "do well in the imitation game"—which Turing also did not define explicitly. However, we believe it is clear from the quality of the Witness's contributions to the imagined dialogue that Turing intended the Witness to succeed *unequivocally*. To quote Harnad [11] (p. 299), the machine's capacities would be "life-size and life long; the candidate must be able to deploy them with anyone, indefinitely", and they must not be "one-night party tricks ... but real, human-scale performance, indistinguishable from our own". To that, we add the notion that conversion includes two participants building a shared understanding, repairing as necessary. Lastly, by *language recognition/transformation*, we mean the ability to recognize human language in some form and transform it to another.

A goal of this paper is to explain the technology behind the "supra-human" achievements recently witnessed by way of giving reasons why we believe it is unlikely a machine will do well at the Imitation Game anytime soon. In our conclusions, we suggest that dialogue is a kind of *practice*, a discipline that might initially be learned by rule-following but eventually transcends the rules, as happens in law, politics, and science. However, everyone engages in dialogue.

## 2. Accidental Philosophers and Hard Problems

Though the computers of his era were crude compared to present-day household gadgets, Turing's 1950 paper has withstood the test of time. Schieber [12] (p. 135) writes, "Turing, in proposing his Test, had packaged in one easily graspable form many of the central problems of philosophy of mind that had exercised people for centuries: the mind–body problem, how mental states relate to the world, the problem of the existence of other minds".

However, Turing's ideas about machine intelligence appear to have shifted in the years immediately preceding 1950. In *Intelligent Machinery* [13], written in 1948 (but not

published until 1968 [14]), Turing discussed "machine intelligence" at length, suggesting that a machine might demonstrate its powers in the following "branches of thought":

1. Various games, e.g., chess, noughts and crosses, bridge, and poker (here we call these "intellectual games");
2. The learning of languages;
3. The translation of languages;
4. Cryptography;
5. Mathematics.

In his 1948 manuscript, he writes that "the learning of languages would be the most impressive, since it is the most human of these activities", although he does not explicitly mention dialog with a human. Turing also sketched out a chess-based variant of the Imitation Game, writing that [13] (p. 431) "It is not difficult to devise a paper machine which will play a not very bad game of chess". He also states [13] (p. 412), "Playing against such a machine gives a definite feeling that one is pitting one's wits against something alive". (Because that era's computers could not run a full-blown chess-playing program, a hidden properly trained human player impersonated (so to speak) a machine by rigidly executing a paper specification of a chess-playing algorithm).

It is striking that Turing's 1950 *Mind* paper makes no use of the expression "branches of thought". Instead, constructive dialogue with a human becomes the paper's centerpiece. Hodges also takes note of this [7] (p. 530), stating that Turing had been careful to choose as the elements of the 1948 list activities "which involved 'no contact with the outside world' [7]. Hodges continues, "The *Mind* paper . . . boldly *extended* the range of 'intelligent machinery' to general conversation (our emphasis)". We will go further and say that Turing's 1950 paper was all the bolder for suggesting that the Imitation Game alone sufficed to *replace* the question "Can machines think?".

Making this shift even more remarkable in hindsight is that significant computational progress has been made in every branch of thought in Turing's original list—but little progress has been made on the Imitation Game.

The next section sketches out recent advances in computer games, the first branch of thought in Turing's list. Many have considered such intellectual games a fine testbed for artificial intelligence because their precise and compact definitions strictly circumscribe the problem domain, yet the games have held a certain intellectual cachet. However, as computers demonstrate the ability to defeat world masters at game after game, it is reasonable to ask whether such performance demonstrates intelligence or just sheer computing power.

Following the section on games, we discuss current language processing technology, which bears similarity to the technology used in computer Go. We characterize the current successes in language processing as language *recognition/transformation*, where recognized language fragments are transformed into another form. This section amplifies observations in the popular press [15,16] that although machines have become more proficient at certain complex language tasks, they have not become more intelligent, largely because they remain highly circumscribed and syntactic and because of their inability to manipulate language according to certain patterns of plausible inference.

Later, we note that Turing's predictions are borne out: significant progress has not been made on unrestricted conversation between machines and humans. Hence, Turing in 1951 observed [17] (p. 489) that this would take "at least 100 years". We further consider conversation as a practice and revisit the idea [18] that is it not yet known whether the Imitation Game is a threshold that we can approach stepwise over time or whether it is a watershed that separates us from machines.

## 3. What Is Easy about Automating Games and What Is Hard?

Many scholars write that early AI researchers believed expert computer performance in games would be best achieved by emulating human experts [19,20]. This idea spilled

into other areas of human expertise such as medical diagnosis [21], where the idea received media attention [22].

We mostly consider games such as tic-tac-toe (called "noughts and crosses" in [13]), checkers, chess, and Go, which are all two-person full information games, that is, games where players take turns and can always see the entire board. An old saying goes that a good game is easy to learn but hard to master. For all these games, the setup is straightforward, the rule sets are small (also unambiguous and easily translated to computer code), but in most cases, they generate a combinatorial explosion of board positions that makes them difficult to master. This explosion is finite, making it possible in theory for an agent to consider in finite time every possible outcome of every possible move and guarantee the best possible result for the agent. Whereas "finite time" might be tractable for a very small game (e.g., $3 \times 3$ checkers), for larger games, it may mean many lifetimes. To manage this, scholars over the years have investigated at least three general approaches to gaming—imitation of human play, efficient exhaustive search algorithms, and statistical search methods—which may be combined in various ways.

### 3.1. Imitating Humans at Games

Though many consider tic-tac-toe child's play, luminaries no less distinguished than Alan Newell and Herbert Simon [20] have provided a complete expert strategy for games similar to tic-tac-toe by way of "cloning" the behavior of experts. Early computer chess programs combined book openings and infallible endgame algorithms with the widely-used technique of hand-crafted numerical evaluation functions [23] (p. 137) that measured the "goodness" of the positions resulting from each possible move, from which the player selects the "best" (for example, a simple chess evaluation function might be the sum of the values of the remaining pieces). The book openings are time-tested, and the useful endgame strategies mostly infallible, but evaluation functions were heuristics, possibly designed by capturing expert instincts in an era when it was not possible for computers to look far ahead.

### 3.2. Exhaustive Search

Exhaustive search proceeds from a current board position, plays all possible subsequent games, then chooses a move that guarantees the computer player a best outcome against *any* player, including an equally powerful machine. *Minimax* [23], an early general-purpose search algorithm for two-person games, traverses the entire search space and suggests a move. *Alpha–beta search* [24] improves on minimax by using earlier search results to ignore branches that cannot guarantee better results than those already found, but its worst-case performance is no better. It has an Achilles heel that we will return to: because these algorithms guarantee an optimal result against an equally strong player, alpha–beta and minimax will choose a move that guarantees a draw even when there is a possibility of winning against a weaker player.

Tic-tac-toe (included in [13]), with only a quarter million board positions, is amenable to this kind of search and remains useful as a testbed for undergraduate AI. As recently as 30 years ago, it was impractical to implement real-time tic-tac-toe with minimax alone. Alpha–beta search made real-time tic-tac-toe almost instantaneous on the computers available then and also motivated the search for faster exact algorithms.

To illustrate how far hardware has come, we implemented bare-bones tic-tac-toe in a few hours using the Python programming language and minimax search with no special features on a recent MacBook. The game hesitated just perceptibly when deciding an opening move, but otherwise appeared to choose moves instantaneously.

In 2007, Schaeffer [19] demonstrated that checkers, with a state space of about $10^{20}$ positions, is *weakly solved*, meaning that the result (at least a draw) and a strategy for achieving it from the beginning of the game are known. To do this, Schaeffer ran as many as 200 processors almost continuously from 1989 until (about) 2007, one of the longest-running computations ever. From this, one can reasonably infer that checkers, like

tic-tac-toe, is deterministic and fully solvable given sufficient resources. Thus, Schaeffer [19] (p. 1518) writes "Perhaps the biggest contribution of applying AI technology to developing game-playing programs was the realization that a search-intensive ('brute-force') approach could produce high-quality performance using minimal *application-dependent knowledge*" (our italics).

For real-time play, even alpha–beta search is not practical. It remains useful because a program using alpha–beta search can look ahead twice as far as it can using minimax in the same amount of time, which makes alpha–beta useful for providing more lookahead in real-time games, at which point evaluation functions kick in.

*3.3. Probabilistic Search*

Schaeffer's weak solvability result makes it difficult to deny that chess and even Go may be solvable, but given the sizes of their state spaces ($10^{50}$ states for chess and $10^{150}$ for Go), weak solutions are unlikely anytime soon, let alone real-time exhaustive search solutions [19]. However, just two years after a Go master was defeated by Google's AlphaGo, its successor, AlphaZero [25], attained "superhuman" performance not only at Go but also at chess and shogi (sometimes called Japanese chess), using what we would call a probabilistic technique.

What is new about AlphaZero is that it learns games entirely from first principles using extensive unsupervised self-play to build a convolutional neural net (CNN) (initialized with random parameters) that for any board position returns an expected outcome for the current player. For chess, this would be a number between $-1$ and $1$, where $-1$ means loss, $0$ means draw, and $1$ means win.

During play, AlphaZero builds a Monte Carlo Search Tree. Suppose player A is at the starting board position *pstart* (the initial root of the search tree). For each legal move $a_i$ available at *pstart*, it retrieves a value $z_i$ from the CNN for the position $p_{a\_i}$ (the result of moving to $a_i$ from *pstart*), giving player A a set of moves and expected outcomes to choose from, and the $p_{a\_i}$ positions become new leaves/children of *pstart* in the tree. If Player A specifically chooses $p_{a\_2}$, a simulated game is played to the conclusion, and all positions along the path from the root to the end node, including $p_{a\_2}$, are updated to include the new information. For the rest of this turn, Player A may play as many simulated games as there is time for, but can now choose between explored and unexplored nodes.

Because of the number of board positions in chess and Go, it is practically impossible to physically store all positions and outcomes. The CNN "statistically shmushes" this information by updating its parameters, which allows it to provide an outcome $z$ for an arbitrary (and possibly unseen) position $p$. The user must accept the (possibly mind-bending) notion that a phenomenal number of board positions and outcomes can be stored in this way, that the information thus stored can be meaningfully interpolated, and that updates will not result in significant loss of information obtained earlier. We remark that some observers remain skeptical about exactly what is stored in neural nets' parameters [26–28], but the fact is that they perform well on these well-defined problems.

Rather than saying the CNN "stores" all this information, a better metaphor might be that it continuously rewrites its evaluation functions as it gains experience, "compressing" each experience into these evaluation functions, and possibly losing some information while improving overall play. Thus, AlphaZero is more adventurous than alpha–beta—it may choose the move with the best expected outcome.

Bridge and poker, two games Turing names in 1948 [13] but not in *Mind* [1], merit different kinds of mention. They differ from the games discussed above in that they introduce uncertainty: cards are shuffled (the introduction of randomness), players cannot see other players' hands (incomplete information), and both games have bidding processes that introduce still more uncertainty, notably bluffing in poker. Nonetheless, they form a closed system where, in theory, every combination of card shuffle and every possible bid could be simulated with exhaustive search and have regularities that could be exploited probabilistically using methods similar to the preceding methods. Google's PoG (Player of

Games) already plays high-level poker [29], and at this writing, NukkAI has defeated eight world bridge champions [30,31].

This kind of uncertainty differs from the kind of uncertainty associated with subjective judgment in some physical sports (e.g., the infield fly in baseball or charging versus blocking in basketball) or that of real-life unanticipated events such as housing bubble bursts, pandemics, and unexpected stock market volatility. We will not be surprised when AlphaZero (or a successor) succeeds at winning all games of real-time bridge, poker, and Go.

## 4. What Is Easy about Language Recognition/Transformation and What Is Hard?

The proliferation of successful tools for computational linguistic tasks (e.g., auto-correct, auto-complete, grammar checkers, speech-to-text transformation, question answering, automatic subtitling, computer translation, optical character recognition (OCR), and voice assistants) may lead some to believe that strong performance at the Imitation Game may come sooner than has been expected. Here, we give our perspectives as to why certain software tools for these tasks have improved, although at least some of the consumer tools still rely heavily upon the ability of the humans interpreting the outputs to "auto-correct" their not infrequent errors. In the end, we conjecture that there is a long way to go before computers can engage in creative constructive conversation.

We begin by considering the use of context in language recognition/transformation, with a focus on language translation. We first consider simple language models by way of introducing the initial successes of the use of context as well as the challenges that remain; we then move on to transformer technology, which is the state of the art at the time of writing.

### 4.1. Early Thoughts on Machine Translation

Like games, languages are easy to learn but hard to master. One problem is that language can be ambiguous even to a native speaker. As early as 1949, Weaver [32] (p. 19) noticed that that "insofar as written language is an expression of logical character, . . . the problem [of translation] is at least formally solvable". Observing that word-by-word translation did not work well, he added that meaning could often be disambiguated using the context of $N$ adjacent words, where $N$ might vary. Some words ('the', 'they', 'when') require no context, others may be disambiguated with some small $N$, but others might require "the whole book", a next-to-impossible technical challenge at the time.

### 4.2. Exploiting Context with Conditional Probabilities

Early purely logical (i.e., grammar-based) approaches to computational linguistics did not yield significant practical results. However, the area gained traction after adopting statistical approaches.

We begin with early work on part-of-speech (POS) tagging in English. We remark here that POS tagging was considered a useful first step for larger recognition/transformation tasks such as translation. To give the flavor of the idea, if POS tags could be attached to words in an English language fragment, it would become easier to identify sequence of higher-level structures, for example, subject, indirect object, verb, and direct object. These might then be mapped into a different higher-level sequence for another language, after which individual words could be translated. This oversimplifies the translation task, but at the very least, given a correct tagging, 'good' the noun could be distinguished from 'good' the adjective.

This effort was bootstrapped by teams of experts (linguists, statisticians, computer scientists) who manually tagged a significant corpus of fragments of text in context (extracts from articles, books, newspapers, and scientific papers). One example is the Lancaster–Oslo–Bergen (LOB) corpus [33], which contained one million words in context, each tagged with one of 150 tags (the LOB corpus was finer-grained than, say, the Penn TreeBank with

36 tags plus punctuation marks). For testing taggers, the corpus might be divided into a 900,000-word training corpus and a 100,000-word test corpus.

A simple tagger may be constructed by first computing two kinds of probabilities from the corpus. The first kind is the probability that a particular tag might display a particular word; the second is the probability that one tag in a sequence is preceded by another. Given a language with, say, 50,000 words, one might fear that this creates too many word/tag and tag/tag parameters. However, many of the parameters are zeroes.

Next, given an input word sequence $w_1 \ldots w_n$ and a set of provisional partial tag sequences, each of the form $t_1 \ldots t_k$, each with score $s$ (initially 1), a bare-bones tagging algorithm could work as follows. Upon seeing word $w_{k+1}$, $s$ is multiplied by the probabilities $p(w_{k+1} \mid t_j)$ and $p(t_j \mid t_k)$, where $t_j$ ranges over all possible tags that might generate $w_{k+1}$, and $t_k$ is the last tag of the partial sequence. The algorithm proceeds until all possible tags are assigned to the last input $w_n$ and the "winning" tag sequence is then the one with the highest score overall.

The astute reader will notice that this calculation is proportionate to the probability that the winning tag sequence generated the observed words, which is the essence of the Hidden Markov Model (HMM) [34]. This compact description omits many details for reasons of space, but to understand how it works in real time, consider the following two sentences.

> She was a good human.

> She was a good human being.

The word 'good' can be a noun (NN), adjective (JJ), or adverb (RB) (as in "she looked good as dead"), with JJ generating the word between 50–100 times more often (in the LOB) than the other two tags, while the word 'human' is a hundred times more likely to be generated by JJ than NN. Finally, 'being' is 20 times more likely to be BEG (a special tag for the verb form of 'being') than NN. Because the words 'She', 'was', and 'a' belong to a special category of *closed-class* words, word that only take on single tags, there is only one possible initial tag sequence for the first three words: PPP3A-BEDZ-AT. A tagger as described above chooses the intuitively correct tag sequence JJ-NN for the last two words of the first sentence but also chooses the intuitively correct sequence JJ-JJ-NN for the second sentence. That is, the tagger resolves the ambiguity of the tag of the word 'good's tag with this small bit of context.

Overall, the tagging accuracy of such a tagger with a few additional heuristics is about 95–97%, a rate that was impressive in its day. (To minimize the number of parameters for use on an inexpensive computer, Neufeld and Adams [34] replaced each word in the corpus with just its last character, reducing the vocabulary to 26 letters of the alphabet plus some punctuation. That tagger trained and tagged on each symbol's last letter, achieving an accuracy of about 70%, a surprise at the time.) A close look at results (correct and incorrect) from this tagger reinforces two notions: firstly, that language may be full of regularities to exploit, but secondly, that this tagger is not "smart", but rather, a high performer probabilistically.

To understand why, suppose 'good' is changed to 'kind' in both sentences. The resulting sentences do not differ much from the originals, but in this case, the successful tag sequences for the two new sentences have the same initial tags but end with NN-JJ and NN-JJ-BEG, respectively, instead of JJ-NN and JJ-JJ-BEG. Since the tagger performs so well generally, and because the two new sentences seem much like the original pair, this is perhaps troubling. The cause is that *kind* is so overwhelming likely to be NN that it throws the rest of the calculation off the rails, and the error propagates.

The problem is akin to what we have called the Watson–Toronto effect [18]. Watson was constructed by IBM to compete in the television trivia game *Jeopardy* against human champions. Its performance astonished viewers until it was given (in the category *U.S. Cities*) the clue, "Its largest airport is named for a World War II hero; its second largest, for a World War II battle". Watson answered "Toronto". To North American audiences, this was

a hysterical blooper, as Toronto is one of Canada's largest cities. To be fair, Watson "knew" it was guessing, and despite this error, it soundly defeated the humans.

### 4.3. Extending Context Flexibly with Transformers

Research on POS tagging has explored many options for improving performance: larger training datasets, additional annotations, and wider contexts. Fast-forwarding to 2017 [35], the transformer, the present standard for language recognition/transformation, eliminated the need for an entire category of neural nets. As with games, this advance was facilitated by using a neural net as a pseudo-database, as well as translating entire phrases, sentences, and paragraphs at a time.

The training phase for a translation transformer requires the collection of a considerable number of input–output (source language/target language) pairs. This distinguishes translation from game-playing—no set of rules exists (yet) that can generate all possible legal input sequences, let alone acceptable input–output pairs, so training cannot rely on the equivalent of self-play.

Testing begins with new source language text where words are embedded into vectors of real numbers, where entries correspond to word features, and words whose embeddings are nearby in this multi-dimensional space are considered similar. As with the interpolation of encoded board positions, the idea of similar words being nearby is also mind-bending, but the proof of the pudding is in the eating: the software performance is improving.

The goal of translation is to find the "best" vector of output words in the target language to match the input, with a presumption that not all inputs have been seen such that the answer can be looked up in a phrasebook. The transformer architecture has the potential of doing something like what Weaver called "reading the whole book", which it accomplishes by using multiples kinds of *attention*, that is, measures of relationships between words and sequences of words.

The *encoder* in the transformer first enriches the initial input with information about the relationships amongst all the words in the sequence, so that no relationships are lost, regardless of relative positions in the input sequence, which it feeds to the *decoder*, which finds the best output sequence by measuring the relationships within the output sequence thus far, as well as the relationships between the initial input and the output sequence thus far as the output sequence is built.

This can be seen in action by using Google Translate to translate the following English sentences to French:

From the bank by the river, I took some ferns.

From the bank by the river, I took some money.

These are rendered as:

Au bord de la rivière, j'ai pris des fougères.

De la banque au bord de la rivière, j'ai pris de l'argent.

(The preceding translation was performed on 1 June 2022 but does not work if you break the input into two sentences. For efficiency, it might be restricting context to a single sentence.)

This section and the preceding skip details. For instance, the LOB corpus happens not to contain the word 'feisty', so one category of solution from that era considers methods for estimating values for unseen parameters for words and tag sequences not appearing in the training data. Our presentation of transformers has not included numerous engineering and mathematical techniques, including how positional information is represented and teacher-forcing.

One key takeaway is that these technologies have enjoyed ever-increasing empirical success by training on increasingly large bodies of language, finding ever more features of language and finding smart ways to store them and ways to approximate given previously unseen inputs. Moreover, this has been done without ruleset games such as chess have that

can be used to determine unequivocal success or failure. However, because transformers use an essentially probabilistic technique to predict outputs, it is implicit in the design that they will still make errors. Thus, we are not aware of any multilingual jurisdiction that has relied entirely on machines to translate its statutes, for example.

### 4.4. Other Language Recognition/Transformation Problems and the Other "Areas of Thought"

Translation offers a useful viewpoint on other language recognition/transformation tasks. In speech-to-text (widely used for real-time subtitling and text dictation), spoken language plays the role of the source language, and text acts as the target. This works because speech can be modeled with a relatively small number of indivisible units of sound called *phonemes*, which combine into *morphemes* (e.g., "un-", "break-", "able-"), which in turn combine into higher forms. The same can be said of optical character recognition technology. Question answering [36] could be viewed as training with questions as the source language and answers as the target, which yields the performance we expect from voice assistants. Text prediction [37] could train a transformer on pairs of language sequence that follow each other with regularity.

We chose to concentrate on translation to give a reader unfamiliar with the recent technology a sense of how language sequences may be arithmetized into multi-dimensional objects in a source format and then may be matched with nearby arithmetized objects in a target format, and that, when this can be done, this technique may be applied in many settings, given sufficient resources. Transformer technology is new, and we expect to see many reports of its successful application in the near future.

What remains interesting is that Turing abandoned his five branches of thought listed earlier for the Imitation Game. In 1948, he stated that games and cryptography required little contact with the outside word, and the same was true to a lesser extent of translation and mathematics, leaving only the learning of languages as depending "rather too much on sense organs and locomotion to be feasible". One could debate whether the Imitation Game is a restatement or extension of "language learning", but what we consider pivotal is that he dropped the list and stated that his Imitation Game, though not necessary, was sufficient.

## 5. How Conversation Differs

### 5.1. An Empirical Argument

Perhaps the state of the art gives some insight into the relative difficulty of these problems. Section 3 explained the current technology behind AlphaZero, a champion at both chess and Go. To sum up, AlphaZero trains by playing against itself using barely more than the chess ruleset, taking only hours to learn to play at championship levels, storing likely-to-win moves in a CNN, which acts as pseudo-database. In play, AlphaZero uses the clock to explore as many new sequences as possible when unseen board positions arise and adds them to its "pseudo-database". For an interesting class of games, the framework is sufficiently general that a champion can be created simply by loading up a new ruleset. While we cannot say that this is true for every game in the world, it is safe to say that computers will be playing many of these games in a league of their own and that many records will be set. It could even be said that, resource requirements aside, AlphaZero handles chess much the way it would handle tic-tac-toe.

Language recognition/transformation can be implemented with different configurations of neural and related architectures such as transformers. Depending on the kind of problem, the transformer is trained on a vast set of relevant input–output pairs that arithmetizes inputs and outputs, so that given new inputs, it can predict likely outputs. Its architecture maintains consistency within the input fragment, within the output fragment, and between the input and output fragments. A major difference between games and language recognition/transformation is that we are not aware of anything analogous to "self-play" for language. That is, there is no relatively compact ruleset that allows a

machine to independently build its own pseudo-database. For this reason alone, we claim that language is different from chess, and this is reflected in the state of the art.

For both games and language recognition/transformation tasks, it is possible (over time, almost certain) that the underlying algorithms, basically probabilistic, will make errors for the same reasons technologies did—they remain quasi-statistical predictions. Present-day language technology might not go off the rails if 'kind' is substituted for 'good' (as in Section 4.2), but the errors made by current language technologies remain a trope. We expect this technology to continue to improve, even at the consumer level.

By comparison, progress in machine–human dialogue, as Turing envisioned it, lags. No machine has been declared (by scientific consensus) to have performed well at the Imitation Game or, more generally, demonstrated an ability to engage in constructive dialogue.

To the contrary, however, a convincing case might be made that participation in conversation might be an extension or variation of question answering and/or language prediction (using tools like GPT-3 [37]) or some hybrid of the two and thus could be trained on a vast database of conversations so that upon receiving a fragment of conversation, it might deliver the best possible response and become increasing accurate over time. Of course, we cannot say that this is impossible. We also cannot exclude the possibility of a new technology, perhaps even one that transcends the Turing machine model, that enables human–machine conversation.

### 5.2. Watershed versus Threshold

Conversational software can be good and useful without doing well at the Imitation Game. Even if there are occasional errors, even serious ones, they might be adopted for the same kinds of reasons as seatbelts and vaccines are: they do more good than harm.

However, in terms of acting as a proxy for intelligence, perhaps it is worth a few words to repeat what we consider the salient features of conversation. We might build a machine that makes mundane statements in an elevator or at a dinner table and is not outed immediately as a machine, but Turing's conversational snippets show he had more than this in mind. The machine would be able to actually engage in conversation, reason about what is being discussed using any number of a possible set of patterns of plausible inference, and contribute productively to the conversation, while pro-actively repairing errors as needed. Such a machine need not be all-knowing or "superintelligent" but able to pass as a human in conversation.

#### 5.2.1. Significance of the Transition between 1948 and 1950

Though the ideas of complexity classes [38] had not entered the lexicon of the day, he references the shortcomings of his era's machines with respect to time. In response to a statement about the possibly millions of years it might take a machine to find a best chess play, Turing states "To my mind this time factor is the one question which will involve all the real technical difficulty" [15] (p. 503).

It is possible that Turing contemplated the idea that in theory a machine might be able to store all possible board positions for chess and look up the best possible move, something AlphaZero approaches. In addition, he had communicated since 1943 with Claude Shannon [7] (p. 314), who later published what is called the Shannon–McCarthy Objection to the Turing Test [39] (pp. v, vi) quoted in [40] (p. 437), which argues that a very large but finite lookup table could generate appropriate responses in dialogue. Turing obviously would have disagreed with this, but he might also have contemplated that a massive phrasebook sufficient for practical translation could be constructed. This may account for the fact that translation does not appear in his *Mind* paper, and chess is not mentioned as a proxy for intelligence.

We wonder if Turing had an inkling that conversation demanded something the other branches of did not.

### 5.2.2. Patterns of Plausible Inference in Language

Recent reports of natural language generators suggest they are far more fluent than Eliza [41], but some observers have suggested they do not appear to be much smarter [42]. This may be in part because much language uses "shorthand" constructs that might be called supra-logical, including implied communication, reasoning about generalizations with implicit exceptions, and reasoning about reasoning [18].

The classic AI example of a failure of implied communication is a robotic agent that replies "Yes" to "Can you pass the salt?" In that light, consider the first few lines of one of Turing's examples [1] (p. 446):

> Interrogator: In the first line of your sonnet which reads 'Shall I compare thee to a summer's day', would not 'a spring day' do as well or better?
>
> Witness: It wouldn't scan.

The interrogator's opening sentence is a yes/no question, yet the Witness does not directly answer and instead heads straight into the explanation—"It wouldn't scan"—which implies that the answer to the question is "no". The Interrogator should make this inference, as part of building their shared understanding as they continue.

In addition, Turing's Witness understands generalizations with exceptions—winter's days are bleak, but Christmas is not. This is also implied knowledge (for a fuller discussion, see [18]). These patterns of plausible inference do not appear to be part of the AI discourse in Turing's era, yet Turing [13] (p. 438) seems to be aware of some of this hidden quality when he writes, "We seem to be quite content that things should not obey too mathematically regular rules. By long experience we can pick up and apply the most complicated rules without being able to enunciate them at all".

There are two issues at play here. One is the knowledge issue, where the Witness understands that winter is bleak, but Christmas is not; that is, the Witness understands the significance of Christmas as something joyous. However, the reasoning problem is that superficially, the sentences are logically contradictory. Yet, the Witness handles both issues with facility.

Lastly, Turing's witness can reason about reasoning, as when the Witness tells the interrogator "I don't think you're serious". That is, the Witness offers an assessment of the Interrogator's entire argument; it is a meta-statement. If, in some other setting, the Interrogator replied, "I *am* serious", together, the parties would undertake to repair their shared understanding.

We contend that the ubiquity of these forms of plausible reasoning contribute to making conversation qualitatively significantly different from chess as well as translation. A human player may think about chess in terms of making an exception to a general rules or reasoning about the reasoning of the opponent, but that is not the case with AlphaZero, which proceeds by selecting the best move given the current board. In conversation, however, the Witness may need to decide whether the Interrogator has made an error or is simply using generalization with exceptions, and whether it can be ignored safely or whether the parties need to stop and clarify. In the transformer architecture, the software just matches an input with the best output.

We remark that a set of papers on such patterns of plausible reasoning appeared as a Special Issue of *Artificial Intelligence* in 1980 [43] on nonmonotonic reasoning, alternately known as common-sense reasoning, that "launched a thousand ships". For at least a decade, this was a vigorous subarea of Knowledge Representation. More recently, this idea has been making a comeback in the context of language understanding [44].

### 5.2.3. Conversation as a Practice

The preceding features are consistent with our understanding of constructive conversation as a social practice. Lay people and philosophers alike have observed that certain skills are learned through a kind of apprenticeship that might begin with rule following then typically continues until the practitioner goes beyond mere syntactic rule-following. One

writer [45] describes the endpoint of this process as "the higher-order skills of responding to the complex demands of a rapidly evolving situation—in a word, *improvising*".

Knight [46] (p. 317) describes a social practice as follows: "to participate in the sharing not only of rules but also of goods, and therefore of reasons for action and—potentially, at least—of cooperative reasoning about action", which bears considerable similarity to our idea of dialogue as "collaboratively building a shared understanding". Knight goes on to say, "even though a practice cannot be sustained if its participants do not normally act in accordance with its rules, those rules may be broken or changed in a way that advances the practice rather than transforms it into a different practice".

Legal systems (but also moral codes) demonstrate humanity's attempt to codify rules of conduct. However, such codes are never complete, since no two cases are identical. At the very least, they occur in different places or times. Thus, people seek recourse from a tribunal. In modern times, this might be a judge who decides the case, possibly setting a precedent, possibly being overruled by a higher court. There is also a relevant discussion about juries in a review [47] of Abramson [48], who distinguishes the representative and deliberative conception of juries. The former view considers the jury simply a representative body of peers. However, a reading of [47] suggests that juries also infuse the practice of law with a collaborative constructed shared understanding of justice. Burns [47] (pp. 1478, 1479) quotes Abramson [48] (p. 8):

> I will argue for . . . a vision that defends the jury as a deliberative rather than a representative body. Deliberation is a lost virtue in modern democracies; only the jury still regularly calls upon ordinary citizens to engage each other in a face-to-face process of debate. No group can win that debate simply by outvoting others; under the traditional requirement of unanimity, power flows to arguments that persuade across group lines and speak to a justice common to persons drawn from different walks of life. By history and design, the jury is centrally about getting persons to bracket or transcend starting loyalties.

There is much to argue with here, but even persons less idealistic about juries might agree that the law should not be blind rule-following, but intelligent and ever-evolving with respect to matters great and small. For an example of a small matter, the reader may wish to review the stormy controversy about whether wall-mounted televisions are fixtures or chattel in real-estate transactions and therefore included (or not) in the sale [49].

This concept of a practice suggests there are social human institutions and activities besides conversation that, while displaying significant regularities, resist capture by a set of rules. From a computer science perspective, this suggests conversation may belong to a class of problems distinct from optimization and enumeration problems. We add that language use is so embedded in other practices that even other language problems, for example, translation of canonical texts, novels, and even laws, especially between cultures, may take on a qualitatively greater dimensionality.

Thus, we regard conversation, the collaborative construction of shared understandings, in this sense is member of the class of practices.

These considerations ground the qualitative belief that the Imitation Game defines a watershed.

## 6. Conclusions

Language processing software can be good and useful without doing well at the Imitation Game. It is also difficult to categorically deny that the capabilities of existing technologies might be incrementally improved and extended until their performance is practically indistinguishable from that of humans and, alternately, that altogether new advances may facilitate this goal.

Nor, on the other hand, does it seem unreasonable to believe that a watershed separates human and machine performance. Quantitatively, we have argued that the state of the art is such that a certain class of games appears to have been mastered by machines, and machines are performing many language transformation/recognition tasks so as to be

useful for wide deployment, while machines have not yet demonstrated significant ability at constructive conversation as we understand it. Turing himself seems to have anticipated at least the relative complexity of language when he defined success at the Imitation Game in 1950 as a proxy for the question "can machines think?" and set aside tasks such as chess-playing and translation. However, again, one might argue that Turing's reference to time constituting the "real technical difficulty" might suggest he considered *only* time to be the problem, and it might be a threshold after all.

We have argued that these quantitative differences may reflect that the Imitation Game is qualitatively different in that ordinary conversation entails patterns of plausible inference and may fall into the category of a practice, both of which present problems for automation.

Whether machines ever converse with us, Turing's vision was far-sighted and wisely cautious and has also been borne out by experience: unequivocal success at the Imitation Game remains, if not the hardest, one of the hardest computational problems in the world.

**Author Contributions:** Conceptualization, S.F. and E.N., software: E.N., investigation, S.F. and E.N., writing—original draft preparation, S.F. and E.N.; writing—reviewing and editing, S.F. and E.N. All authors have read and agreed to the published version of the manuscript.

**Funding:** Internal funding, University of Saskatchewan.

**Informed Consent Statement:** Not applicable.

**Data Availability Statement:** Not applicable.

**Acknowledgments:** Thanks to several anonymous reviewers for careful remarks of as well as the the editors of *Philosophies* for their meticulous work.

**Conflicts of Interest:** The authors declare no conflict of interest.

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
