# Peer review of "The Accidental Philosopher and One of the Hardest Problems in the World"

_philosophies, doi:10.3390/philosophies7040076_

Round 1
Reviewer 1 Report
This paper is an interesting contribution to the discussion over the Turing Test. The author suggest that unconstrained conversation is an excellent proxy for the question of whether machines can think. But the paper should be improved in some ways:
- It is an overstatement that solving the Turing Test is the "hardest problem in the world. Even in the conclusion, it's stated that it is merely one of such problems. Please be more reasonable. Solving world hunger, attaining eternal peace, solving all health problems and conquering death – these seem much harder. Not to mention solving NP-complete problems, which are likely formally computationally harder than human conversation.
- The paper suffers from some remnants of the AMRAD structure -- there is a 'Discussion' section that does not seem in any way special as compared to previous ones. Please rename.
- It's unclear to me why recent work on question-answering (QA) in NLP is not even mentioned once, while there have been important new developments there. I also don't understand why CNNs are mentioned while most progress comes from the transformer architecture.
- The paper does not seem to offer a new insight on what makes natural conversation so difficult -- in more exact terms. Is any human practice so difficult? Or any human coordinated practice? (see the last minor issue below)
Minor issues:
p. 1, line 34: it's usually 'the Turing Test' (with the definite article) but your usage is inconsistent (sometimes you drop the article for unclear reasons)
p. 6 your example of POS tagging is somewhat misleading because English is morphosyntactically simpler than most human languages (it's morphology is much easier to account for computationally than Hungarian or Finnish). I would be surprised to see a Hungarian tagger that is based merely on word frequency and sequences of two subsequent POS tags and has over 70% F1 score. This simply doesn't work for compounding languages.
p. 6, line 256: the most popular tagset (Penn Treebank tagset) for English has 36 tags, not 150 - you could have more but this is not typical for English. LOB corpus was not originally tagged, still the official download from the Oxford Text Archive features a tagged version in the Penn tagset. Where's this 150 coming from? (same on line 298, p. 6)
p. 6, line 271: actually, linguists usually think that nouns can form noun phrases without becoming adjectives; if you search for "work" day in British National Corpus, "work" is tagged as NN in phrases like "his last work day". We have zillions of examples when nouns are used without the possessive ending but retain their noun character ("computer network", "network card", "journal article"). I'd drop this point because it does not reflect typical linguistic analyses.
p. 7, line 308: you use the term "text recognition" without any definition. It's not clear to me whether it includes, for example, text summarization, or not, or question answering (more relevant to the Turing test)
p. 7, first paragraph: you underestimate the difficulty of some text processing tasks, such as word sense disambiguation or, particularly, word sense induction, which are rather poor --see http://nlpprogress.com/english/word_sense_disambiguation.html_
You also underestimate the difficulty speech recognition for natural conversation. As far as I remember, Roger Moore stressed that speech recognition for multiple speaker speaking past one another is basically below chance levels.
p. 7, lines 347-352: too many things are packed into a single sentence. It's really convoluted, and I had to read it several times to see that there is a digression there.
p. 8, line 419: "we cannot imagine…": well, I can – just have a huge database with subtitled movies / tv series episodes and perform some training in the way similar to how visual QA is done. You could crowdsource people to tag phone conversation scenes in a soap opera easily, and then classify human actions along with their speech acts. Your scenarios seem pretty typical for phone conversations. What's wrong here with my proposal?
p. 9, line 444: you may want to say that the very first theorem prover by Newell and Simon provided nicer proofs for some theorems in Principia (as Bertrand Russell noticed in his letter) but the paper with these results was very difficult to publish (just a side note)
p. 10, line 472: "there are human institution and activities… that resist capture by a set of rules" Why exactly? Maybe people wanted to capture a complex system in 10 rules, or 100, while there should be something like a billion rules / or a huge overparametrized transformer NN? Is there any mathematical proof that this would be impossible, or is it merely handwaving?
p. 10, line 491: 'there is no obvious analogue to self-play': Vygotsky would deny this. Inner speech in babies and toddlers seems akin to self-play, and one could study what makes inner speech acts successful or not.
Author Response
Hello, thank you for the thorough review. The manuscript was changed quite a bit and our response details the changes.

Reviewer 2 Report
An interesting overview of progress made on language recognition and domain-specific tasks since Turing's famous Mind paper.
My only suggestion is that the discussion of Warren Weaver's famous idea is a little too optimistic. Linguists don't generally think it's appropriate to treat languages as encodings of one another. And at least one cognitive scientist has objected to Weaver's viewpoint, on the basis that contemporary translation software falls easily into rather simple traps: https://www.theatlantic.com/technology/archive/2018/01/the-shallowness-of-google-translate/551570/
Other than that, I found two typos:
- p. 3: "...which is bears some similarity..."
- p.7: "Weaver [[32]" should only have one opening square bracket.
Author Response
Thank you for your review. Attached is a word document with details.

Reviewer 3 Report
Well written and clear survey of the possible Turing tests that he seems to have considered.
But the reader has a hard time extracting a research thesis. Any system with a finite set of objects and finite set of rules for combining them will be computable. Games fall into this category. Conversation, covering endless scenarios, doesn't.
If the intended reader does not already know this, then this will be enlightening and engaging.
Line 34: ...THE Turing Test..." Or do you mean the phrase "Turing Test"?
Line 42: State the error precisely.
Line 49: But this is not exactly the original test, in which the observer distingushed male from female.
Line 114: Sentence fragment?
Line 146: Missing word "designed TO capture"
Line 297: The point is not clear. Whatever use POS tagging provided before, it still provides, right?
Line 437: examples... of what, exactly? Reference has been lost.
Your designation of conversation as a practice is cogent. But is it new? This reviewer would welcome a few words about what that classification adds to the vast commentary on Turing tests.
Author Response
Thank you for your comments. The attached word document contains responses to your concerns.

Round 2
Reviewer 1 Report
The revision addressed my previous comments very well. Small issues that could be addressed are:
1) "at that time", line 9 – it's a bit ambiguous given that you say something about the test of time on the same line; maybe rephrase?
2) "sequence of higher level structures", line 294; "sequences" seems more apt here
3) "where entries", line 374, leave only "entries" (you already have "where" in that statement)
4) reference 27: "Wojciech" is the first name of the author Wojciech Samek, it should be "Samek, W." instead, line 697